

**An instrument for the rapid quantification of PM oxidative potential: the Particle Into Nitroxide Quencher (PINQ)**

Reece A. Brown[1], Svetlana Stevanovic[*1,3], Steven E. Bottle[2], Zoran D. Ristovski[1,2*]

[1] ILAQH (International Laboratory of Air Quality and Health), Queensland University of Technology, 2 George St, Brisbane, 4000 QLD, Australia

[2] School of Chemistry, Physics and Mechanical Engineering, Queensland University of Technology (QUT), Brisbane, 4000 QLD, Australia

[3] School of Engineering, Deakin University, VIC 3216, Australia

*Correspondence to*: Svetlana Stevanovic (s.stevanovic@deakin.edu.au)

**Abstract.** Presence or generation of reactive oxygen species (ROS) on/by particulate matter (PM) have been implicated in PM-induced health effects. Methodologies to quantify ROS concentrations vary widely both in detection and collection
methods. However, there is currently an increasing emphasis on rapid collection and measurement due to observations of short half-live ROS. To address this problem, this manuscript details the design and characterization of a novel instrument named the Particle Into Nitroxide Quencher (PINQ). This instrument combines the 9,10-bis (phenylethynyl) anthracene-nitroxide (BPEAnit) ROS assay in conjunction with a purpose-built aerosol collection device, the insoluble aerosol collector (IAC). The IAC continuously collects PM regardless of size or chemistry directly into a liquid sample with a collection efficiency of >
0.97 and a cut-off size of < 20 nm. The sampling time resolution of the PINQ is one minute, with a limit of detection (LOD) of 0.08 nmol.m$^{-3}$ in equivalent BPEAnit-Me concentration per volume of air. This high sample time resolution and sensitivity is achieved due a combination of the highly concentrated IAC liquid sample and the rapid reaction and stability of the BPEAnit probe.

## 1. Literature Review

**1.1 Introduction**

Atmospheric pollution is responsible for more than 7 million premature deaths each year. A large contributor to this increased mortality is atmospheric particulate matter (PM), which has been linked to: cardiovascular disease (Chow et al., 2006; Donaldson et al., 2001; Nel, 2005); increases in the prevalence of chronic respiratory disorders (Nel, 2005)(Penttinen et al., 2001); and adverse effects in both embryonic and adult neuron activity (Morgan et al., 2011). Furthermore, studies have shown
a link between exposure to PM in diesel exhaust and an increased lung cancer risk (Silverman et al., 2012). A consequence of these and other discoveries was that in 2012, diesel exhaust was classed as a type 1 carcinogen by the International Agency for Research on Cancer (IARC) (Benbrahim-Tallaa et al., 2012).



A key mechanism used to explain these adverse health effects is oxidative stress (Ayres et al., 2008; Li et al., 2002). Reactive oxygen species (ROS) are a group of free radicals which can be either: present on the surface of PM (exogenous ROS)(Hung and Wang, 2001; Venkatachari et al., 2007); or generated through chemical reactions between PM and cells (endogenous ROS)(Donaldson et al., 2002). When inhaled, PM-associated ROS interact with cells to create oxidative stress, interrupting

cell function and potentially leading to cell inflammation and death. The ultrafine particle range have been shown to be particularly hazardous in this respect. Their ability to penetrate deeper into tissues than their larger counterparts allows them to collect inside mitochondria, causing major structural damage to cells (Li et al., 2002).

Aerosols generated through combustion processes are of particular concern for oxidative stress as they are a major source of ultrafine particles (Brines et al., 2015; Posner and Pandis, 2015) and their potential for high ROS content (Cho et al., 2005;

Kao and S., 2002; Mudway et al., 2005; Ristovski et al., 2012; Zhao and Hopke, 2012). Combustion emissions associated with vehicle emissions, power generation and biomass burning significantly contribute to the aerosol burden in many heavily populated urban areas (Harrison and Yin, 2000; Ma et al., 2017; Simoneit et al., 2004); making PM-associated ROS a key issue in the assessment and understanding of the health impacts of air pollution. There have been many in vivo investigations involving both the monitoring of human exposure, and the exposure of animals and lab-cultivated cells to ROS-containing

aerosols (Donaldson et al., 2002; Morgan et al., 2011; Sa et al., 2014; Shi et al., 2006; Shima et al., 2006). Whilst these studies provide us with an understanding of the aforementioned health impacts of PM-associated ROS exposure, they lack the ability to quantify ROS concentrations. In order to achieve ROS quantification, termed herein oxidative potential, a range of different in vitro methodologies have been developed.

**1.2 Chemical Assays**

The measurement of oxidative potential is a complex issue in terms of both sample collection and chemical analysis. In order for measurement to take place, PM-associated ROS must in most cases be collected into a liquid and mixed with a chemical probe. The degree to which this probe reacts with the sample is then measured in order to ascertain a value of oxidative potential. Many different chemical probes have been used for this purpose, all with various benefits and shortcomings. A detailed review of the most commonly used probes can be found elsewhere (Hedayat et al., 2015).

Some probes, including p-hydroxyphenylacetic acid (POHPAA) (Hasson and Paulson, 2003) and ascorbic acid (AA) (Fang et al., 2016; Stoeger et al., 2009), are only sensitive to narrow ranges of ROS species; making them unsuited for quantification of total oxidative potential. The dithiotreitol (DTT) assay is commonly used (Eiguren-Fernandez et al., 2017; Fang et al., 2014; Gao et al., 2017; Li et al., 2002, 2009, Sameenoi et al., 2012, 2013) due to its ability to simulate the reaction responsible for the generation of ROS when PM interact with cells; making it a suitable assay for the quantification of endogenous ROS. DTT

typically requires reaction times of up to 90 minutes with a sample for accurate quantification (Cho et al., 2005), limiting real time applications. However, new developments in microfluidic sensors have more recently allowed for DTT measurements with reaction times as low as 18 minutes (Koehler et al., 2014; Sameenoi et al., 2012).



2,7-dichlorofluorescein diacetate (DCFH-DA) combined with a horse radish peroxide (HRP) catalyst is currently the most commonly used ROS probes in literature for the quantification of exogenous ROS (Fuller et al., 2014; Huang et al., 2016b, 2016a; King and Weber, 2013; Venkatachari et al., 2007; Venkatachari and Hopke, 2008; Wang et al., 2011; Wragg et al., 2016; Zhou et al., 2017). This is due to its relatively simple fluorescence-based quantification, potential for semi-continuous

monitoring and sensitivity to several different ROS species. However, DCFH is prone to several issues, including: auto-oxidation upon exposure to air and sunlight (Stevanovic et al., 2012a); a high background fluorescence and a lack of sensitivity (Pal et al., 2012); a relatively complex chemistry setup for implementation, especially in the case of online measurements (Huang et al., 2016a; Wang et al., 2011; Wragg et al., 2016; Zhou et al., 2017); the catalytic activity of the HRP catalyst is dependent upon sample composition (Pal et al., 2012); and a minimum reaction time of 11 minutes for quantification with

HRP (Zhou et al., 2017), and a 60 minutes without HRP (Pal et al., 2012). These attributes make this probe very difficult to work with in the field, and highlight the need for an alternative ROS probe for accurate oxidative potential measurements. Profluorescent nitroxides (PFNs) are a group of chemicals consisting of a nitroxide group bound to a fluorophore (Fairfull-Smith and Bottle, 2008). Initially these molecules are only weakly fluorescent, however upon reaction with free radicals they become highly fluorescent (Blinco et al., 2011). Quantification of these reactions can be achieved by comparing the

fluorescence intensities before and after reactions. Several different PFNs have been developed at the Queensland University of Technology, one of which is BPEAnit. When placed into solution with dimethyl sulfoxide (DMSO), BPEAnit has been shown to be sensitive to a broad range of exogenous PM-associated ROS (Stevanovic et al., 2012b), in particular those generated in combustion emissions (Miljevic et al., 2010; Stevanovic et al., 2012a). Furthermore, its reaction is diffusion limited, allowing for relatively quick measurement quantification. These characteristics make the BPEAnit a promising assay

for the rapid online measurement of exogenous ROS concentration and oxidative potential.

### 1.3 Measurement Techniques

Beyond chemical probes, there are several properties of ROS and combustion aerosols which complicate oxidative potential measurements. Exogenous ROS react readily with the atmosphere and other surroundings (Fuller et al., 2014). This is a significant issue with standard filter capture techniques, as the oxidative potential can be skewed due to the decay of collected

PM-bound ROS during long periods of collection and storage prior to measurement (Fuller et al., 2014; Zhou et al., 2017). Therefore, methodologies involving either: long periods of collection; or delays between collection and measurement, risk severely underestimating the total oxidative potential of aerosols. Additionally, extraction processes to remove particle from filters for analysis can introduce further positive and negative sample artefacts (Miljevic et al., 2014). To address these issues, methodologies have been developed to rapidly collect PM directly into liquid for more accurate quantification of oxidative

potential.



| Instrument | Chemical Assay | Collection Method | Insoluble Particles | Sample Flowrate | Time Resolution | Limit of Detection |
|---|---|---|---|---|---|---|
| ROS Sampling-Analysis System | DCFH | PILS | No | 16.7 Lpm | 10 min | n/a |
| OPROSI | DCFH | PC | No | 5 Lpm | ≤ 12 min | 4 nmol.m$^3$ |
| GAC-ROS | DCFH | GAC | No | 16.7 Lpm | 20 min | 0.12 nmol.m$^3$ |
| ROS Analyser | DCFH | PC | No | 1.7 Lpm | 8 min | 2 nmol.m$^3$ |
| o-MOCA | DTT | LSS | Yes | 3 Lpm | 3 hr | 0.15 nmol.min$^{-1}$ |
| Online DTT Monitoring System | DTT | PILS | No | 16.7 Lpm | 3 min | n/a |

*Table 1 List of online ROS instruments and their key characteristics including: ROS Sampling-Analysis System (Venkatachari and Hopke, 2008; Wang et al., 2011); OPROSI (Fuller et al., 2014; Wragg et al., 2016); GAC-ROS (Huang et al., 2016a); ROS Analyser (Zhou et al., 2017); o-MOCA (Eiguren-Fernandez et al., 2017); and Online DTT Monitoring System (Koehler et al., 2014; Sameenoi et al., 2012). Limit of detection (LOD) for DCFH systems is given as equivalent $H_2O_2$ concentrations*
*per cubic meter of air. O-MOCA LOD is given as DTT consumption per minute.*

### 1.3.1 Steam Collection Devices

*Table 1* provides a list of the relevant online instruments developed for the quantification of PM oxidative potential. Each instrument uses a different combination of either the DTT assay or DCFH assay with a method for direct capture of PM into liquid. The most predominant of these particle collection methods used are steam collection devices (SCD) (Khlystov, 1995;
Kidwell and Ondov, 2010; Simon and Dasgupta, 1995), including: the Particle Into Liquid Sampler (PILS) (Orsini et al., 2003; Weber et al., 2001) and the Gas Aerosol Collector (GAC) (Dong et al., 2012). SCDs utilize condensational growth to achieve high efficiency collection of ultrafine particles. When in operation, a sample aerosol is continuously mixed with a precisely controlled flow of steam and cooled, creating a supersaturated aerosol. Particles inside this aerosol undergo condensational growth, increasing them to a size typically between 1 and 4 µm in diameter (Orsini et al., 2003). These grown particles are
then collected into liquid with a very high efficiency through wetted impaction methods. The direct injection of steam allows for high supersaturations to be achieved with higher flowrates (Hering et al., 2014). For this reason, steam collection devices can achieve highly concentrated samples, as the ratio of aerosol sample flow to liquid collection flow can be maximized.

Both the PILS and GAC report very efficient collection of PM, with efficiencies of: a collection efficiency of 97% particle number collection efficiency for particles between 0.03 – 10 µm (Orsini et al., 2003); and a total mass collection efficiency of



> 99.5% (Huang et al., 2016a). The distinction between number collection efficiency and mass collection efficiency is important. Whilst the PILS system collects efficiency over all particle modes from fine to coarse, the GAC has a low collection efficiency for particles < 150 nm (Dong et al., 2012). This is due to high ultrafine losses inside a wetted annular denuder (WAD) (Simon and Dasgupta, 1993, 1995) used for the GACs gas phase measurements.

The collection method typically used to capture particles in SCDs involves the continuous deposition and washing off of particles on a solid surface. Hence, this method relies on particles being soluble in the collection liquid for its reported high collection efficiency. An alternative collection method was proposed in a publication focused on developing a system for the collection of virus aerosols. The impactor plate of a PILS was replaced with a wetted-wall PM cyclone which collects particles into a standing liquid vortex (Orsini et al., 2003). This prevents particles from depositing on a solid surface, allowing them to
be captured into the sample liquid for analysis regardless of solubility. None of the instruments for the measurement of oxidative potential discussed here have used this method.

### 1.3.2 Particle Collectors

The particle collector (PC) (Takeuchi et al., 2005) is another particle collection method shared by published oxidative potential devices. Names of similar devices include the mist chamber and aerosol collector. Variants of the PC are used in both the
Online Particle-bound ROS Instrument (OPROSI) and the ROS analyser systems. The PC operates by collecting PM onto a hydrophilic filter which is continuously wetted with a fine mist of a capture solution. The capture solution drips off the filter into a collection reservoir underneath, removing the soluble portion of the collected PM for analysis. The use of a filter allows for the collection of ultrafine PM with a very high efficiency (97.7 % (Takeuchi et al., 2005)) without the need of a condensational growth stage. A disadvantage of the PC system that the collection reservoir limits the sample resolution of the
system due to the continuous mixing of new and old sample.

### 1.3.3 Liquid Spot Sampler

The o-MOCA system (Eiguren-Fernandez et al., 2017) utilizes a Liquid Spot Sampler (LSS) (Hering et al., 2014) to collect particles for analysis. Similar to the SCDs, the LSS system utilizes a condensational growth process to collect ultrafine particles. However, the method of implementation is significantly different. Rather than mixing the sample aerosol with steam,
the aerosol instead undergoes a three-stage water condensational growth process (Hering et al., 2014). The sample aerosol is drawn through a wet-walled tube where it passes through three independently-controlled temperature regions. The induced temperature differentials combined with diffusion of water vapour from the wetted walls of the tube lead to condensational growth of < 10 nm particles (Hering et al., 2014). The grown particles are impacted into a small liquid volume over the period of 3 hours before being pumped off for analysis.
The reliance on diffusion to introduce water vapour to the sample limits the operational flowrate, leading the LSS to use three separate 1 L.min$^{-1}$ growth tubes for a total sample flow of 3 L.min$^{-1}$. The direct impaction into liquid removes the reliance of the PM to be soluble in the collection liquid, allowing for solubility-independent collection.





### 1.3.4 Measurement Technique Comparisons

Ultrafine PM concentrations are heavily dependent upon source proximity and atmospheric conditions (Sabaliauskas et al., 2013) which can lead to significant variations over short time periods and distances. This, coupled with the short half-life of some exogenous ROS species (Fuller et al., 2014; Zhou et al., 2017); indicates that the oxidative potential of ambient PM is dynamic and prone to significant changes over short distances and times. Therefore, in order to accurately measure and understand the health impacts of PM oxidative potential, ROS monitors must have time resolutions sufficient to accurately quantify these variations.

The time resolution of the discussed instruments varies from widely from 3 minutes to 3 hours. It is clear that the SCD and PC based instruments offer much higher potential sample resolutions than that of the LLS-based OPROSI. Within the SCDs, the PILS-based systems have, at a minimum, half the sample resolution of the GAC-ROS due to its dual gas and particle phase measurements. This dual measurement also causes the GAC-ROS system to have a low collection efficiency in the ultrafine PM range, indicating that a SCD without a gas phase measurement stage is a superior choice for PM-associated ROS measurements. The PILS-based Online DTT Monitoring System has the highest sample resolution of all the discussed instruments. This is because the total sample liquid volume of the instrument is much lower as it does not requiring a reservoir as is the case of PC systems. For this reason SCDs currently offer the highest potential for high sampling resolution instruments for oxidative potential measurements.

Aside from the OPROSI, a common disadvantage shared by all current systems is their inability to reliably collect insoluble particles. This is problematic as a significant portion of primary combustion emissions are hydrophobic (Popovicheva et al., 2008). Furthermore, it is not possible to correct for this insoluble fraction in post-analysis of the data using known losses. This is because the insoluble fraction represents a distinct group of PM with both a separate physiological impact (Delfino et al., 2010), and a different oxidative potential (Verma et al., 2012). Future systems should endeavour to adapt or create new methodologies to allow for the collection of insoluble particles to improve the understanding and toxicity of aerosol oxidative capacity.

### 1.4 Manuscript Focus

This paper discusses the design and testing of a novel instrument called the Particle Into Nitroxide Quencher (PINQ). The PINQ has been developed to address the need for an accurate and repeatable method of measuring the oxidative potential of aerosols. Informed from the current literature on the field, the instruments collection mechanism is based on a steam collection device, with a wetted-wall cyclone continuously collecting PM directly into a solution containing the BPEAnit ROS assay. The fluorescence increase of the BPEAnit probe is measured using a purpose-build flow through fluorimeter with a low internal volume and minimal flow-path length to ensure rapid quantification of oxidative potential.



## 2. Instrument Description

### 2.1 PINQ Layout

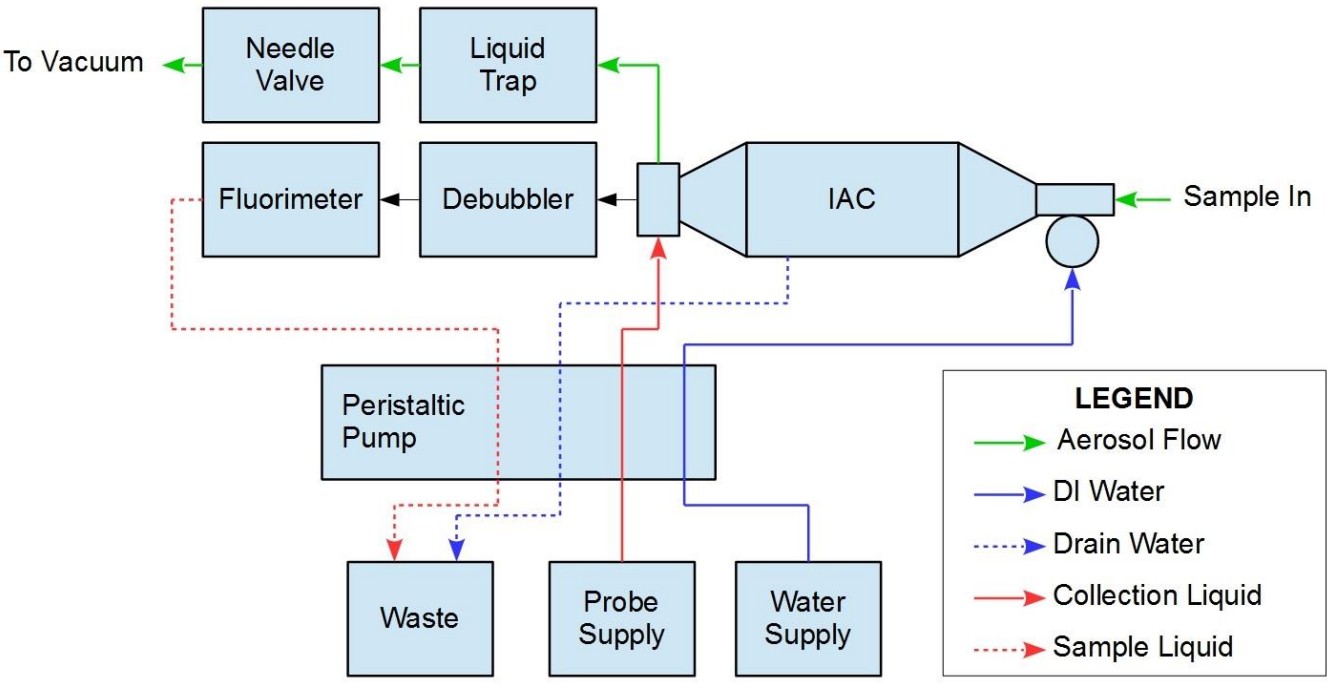

*Figure 1 Diagram of the PINQs Insoluble Aerosol Collector (IAC) showing key components including: the aerosol inlet, steam generator, growth chamber and vortex collector.*

The flow diagram of the PINQ can be seen in *Figure 1*, including all key components of the system and their corresponding connections through liquid and aerosol flows. Aerosol flowrate through the system is regulated via a needle valve connected to a vacuum, whilst liquid flows are controlled using a peristaltic pump. The aerosol is collected in a novel instrument called the Insoluble Aerosol Collector (IAC), in which PM is collected continuously into a liquid sample independent of size or chemical composition. The sample solution is then debubbled and passed through a purpose-build flow-through fluorimeter to quantify oxidative potential.

### 2.2 The Insoluble Aerosol Collector (IAC)

The PINQs aerosol collection stage, the IAC, is categorized as a steam collection device. The sample aerosol is mixed with water vapour to generate a supersaturated aerosol in which the PM underdo condensational growth. These grown particles are collected with a high efficiency into a continuously flowing sample solution inside a solvent resistant vortex collector. The system can be divided into four main sections: inlet; growth chamber; vortex collector; and steam generator.

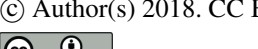



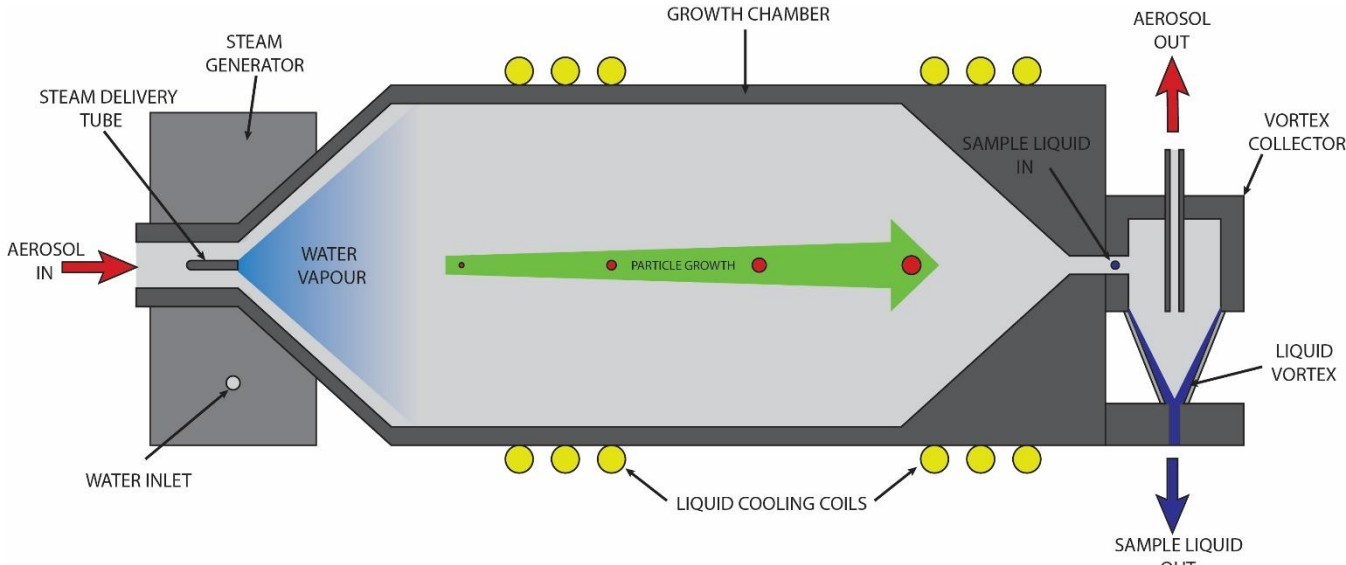

*Figure 2 Diagram of the PINQs Insoluble Aerosol Collector (IAC) showing key components including: the aerosol inlet, steam generator, growth chamber and vortex collector.*

### 2.2.1 Inlet

The IAC inlet contains a fine aluminium steam delivery tube concentric to the inlet, oriented in the direction of flow. A precisely controlled mass flowrate of water vapour is injected into the aerosol flow through this delivery tube. The aerosol and water vapour are then turbulently mixed through an expanding cone, creating a homogenous supersaturated mixture. The standard sample flow rate for the instrument is 16.7 L.min$^{-1}$ (1 m$^3$.hr$^{-1}$) with a corresponding steam generator water supply of 1.5 mL.min$^{-1}$.

### 2.2.2 Growth Chamber

The supersaturated aerosol is passed through an aluminium growth chamber to cool the mixture and allow time for particle condensational growth to occur. The chamber is actively liquid cooled to prevent any excess heat build-up in the system from the steam supply or the aerosol sample. A drain at the base of the chamber removes excess condensation on the chamber walls. Grown particles exiting the growth chamber are accelerated through a contracting cone into the vortex collector.

### 2.2.3 Solvent-Resistant Vortex Collector

The vortex collector is a specially designed aerosol capture device which collects particles into liquid regardless of solubility. The concept was based on a wetted-wall cyclone described previously in literature (Orsini et al., 2008).  It is similar to a miniature PM cyclone in design, with the dimensions of the internal cone and the inlet and outlet tubes selected in order to create a stable air cyclone inside the device. However, whilst the PM cyclone deposits particles onto the solid surface of the





cone, the vortex collector deposits particles directly into a collection liquid. This is achieved through the continuous injection and removal of liquid at the top and base of the cone, respectively. The inertial force of the air cyclone acts on the liquid as it passes through the cone, causing the liquid to spread on the cone wall and form a standing liquid vortex. Grown particles entering into the cyclone from the growth chamber impact into this liquid vortex and are captured directly into the liquid

stream.

In principle the design of a vortex collector is similar to that of a PM cyclone. However, whilst the cyclone design focus is on cut-off size and shape; the vortex collector is predominantly concerned with the formation of a stable liquid vortex. Parameters influencing this include: dimensions; liquid and aerosol flowrates; sample liquid; temperature; materials; and surface finish, making prediction of vortex formation difficult. To this end, the vortex collector was constructed using a combination of

aluminium and stainless steel with a transparent fluorinated ethylene propylene (FEP) cone. This ensures the instrument is solvent resistant for use with DMSO whilst also allowing for visual confirmation of vortex formation.

This device was chosen over simpler impaction methods for two reasons. First, the vortex collector collects particles directly into liquid allowing for solubility-independent particle collection. Second, the liquid vortex ensures that the collected PM is homogenously mixed with the sample liquid exiting the cone base. When integrated into the PINQ, this factor coupled with

the diffusion limited reaction between BPEAnit and collected ROS leads to the sample liquid exiting the base being fully reacted and ready for quantification.

### 2.2.4 Steam Generator

A stable and precisely controlled steam flowrate is essential for effective condensational growth of ultrafine particles. Initially a design similar to that described in literature was used (Orsini et al., 2003), in which a regulated water flow was input into a

heated copper tube. A similar design was attempted with both one and two stage heating, but was fraught with stability issues. This was attributed to the system having a low thermal mass, causing an unstable equilibrium point in the system. Furthermore, the gradual corrosion of the copper tube lead to concerns over potential sample contamination. To remedy these issues, a new generator was proposed with a larger, insulated thermal mass.

 The new generator design consisted of a cylindrical aluminium block sheathed in thermally insulating polytetrafluoroethylene

(PTFE). The aluminium block contains a sealed internal chamber with narrow inlet and exit holes bored perpendicularly at the base and top, respectively. The block is maintained at a constant temperature by the use of two heater cartridges and a thermocouple driven with a Novus N1020 PID controller. A flow of water regulated by a peristaltic pump is continuously input into the base of the chamber through the inlet hole, where it vaporizes and expands, before being ejected through the exit hole. The aluminium steam delivery tube from the inlet section is inserted directly into the exit hole to minimize cooling and

reduce system size. This design greatly improved stability and allows for long periods of unattended sampling.



## 2.3 Flow-Through Fluorimeter

A compact flow-through fluorimeter was designed to provide fast and accurate measurement of sample fluorescence for ROS quantification. The sample liquid fluorescence is continuously measured in a small flow-through quartz cell contained within a stainless steel housing. The excitation source is a 450 nm diode laser, and the fluorescence response measured perpendicularly with an Oceanoptics USB2000+ spectrometer. The laser and spectrometer are directly mounted to the stainless steel housing to remove the need for optical fibre connections, reducing setup complexity and minimizing size. The setup is controlled using a dedicated LabVIEW-based application.

## 3. Methodology

This study is divided into two investigations. The first is the measurement of the IAC collection efficiency. The second is the characterisation of the PINQ system fluorescence response. Flow diagrams of experimental setups, further methodology details, and calculations can be found in the supplementary material.

### 3.1 IAC Characterization

### 3.1.1 Particle Mass Collection Efficiency

The collection efficiency of the IAC for both fine ($PM_{2.5}$) and ultrafine ($PM_{0.1}$) particles was investigated through the comparison of mass concentration of ammonium sulphate (AS) collected with the IAC and those collected onto membrane filters. Sample aerosol was generated using a solution of AS in DI water in a Mesa Labs 6 Jet Collision Nebulizer. Sample was dried and diluted to provide the required sample flowrate before being sampled by the IAC, filter (Whatman Nuclepore polycarbonate membrane, 25 mm diameter, 0.2 µm pore size ) and scanning mobility particle sizer (SMPS) (TSI 3071 Classifier, TSI 3772 Condensation Particle Counter, Aerosol Instrument Manager (AIM) software). IAC samples were collected into sealed quartz sample vials. Sample filters were collect and placed in quartz vials where they were sonicated for 5 minutes in 1 mL of solution. Samples were refrigerated until ion chromatography (IC) analysis was performed by the Central Analytical Research Facility (CARF) based at the Queensland University of Technology (QUT). Measurements were performed using a Dionex Integrion High Precision Ion Chromatography setup with a Dionex AS-AP autosampler. An AS18 Column (150 mm x 2 mm) with an isocratic 0.30 mL/min flow was used. 0.1, 1, 5, 10, 20 and 100 ppm standards were used as calibration points.

### 3.1.2 Size Dependent Particle Number Collection Efficiency

The size dependent particle number collection efficiency was investigated using Di-Ethyl-Hexyl-Sebacat (DEHS) particles. This was chosen as the droplet activation diameter is not dependent on the composition of the particles if they are hydrophobic (Andreae and Rosenfeld, 2008). Particles were generated using a Mesa Labs 6 Jet Collision Nebulizer containing a solution of





DEHS in ethanol.. Monodisperse particle sizes were size-selected from the resulting polydisperse aerosol using a TSI 3071 Electrostatic Classifier for sizes between 30 and 300 nm. The aerosol was then diluted to provide sufficient sample aerosol flowrate. A 3-way valve was used to rapidly switch between pre and post instrument sampling, with particle concentrations for each size found by averaging a 60 s sample from a TSI 3025 butanol-based CPC after system had stabilized. A nafion dryer
was attached to the CPC inlet to prevent the high humidity of the sample influencing measurements.

### 3.2 PINQ Characterization

The PINQ was assembled for characterization as per the layout described in *Figure 1*. A 1 µM solution of BPEAnit in DMSO was prepared as the sample solution for the PINQ. A 1.5 mL.min$^{-1}$ steam generator water feed rate and a 1 mL.min$^{-1}$ sample flowrate was used. The PINQ inlet was connected to a three way valve, with one port connected to a continuous supply of high
purity nitrogen gas and the other connected to a combustion sample chamber. A cigarette was left to smoulder in the sample chamber for 5 min before being extinguished. The PINQ then sampled continuously, with the supply switched between nitrogen gas and cigarette smoke every minute for a total of eight samples.

The fluorescent response of the BPEAnit probe is expressed as an equivalent nanomolar increase in the concentration of the methyl adduct of the probe (BPEAnit-Me). The conversion factor for this is found through the slope of a calibration curve
generated from fluorescence measurements of BPEAnit-Me standards of various concentrations. For PINQ measurements this equivalent response is then normalized against the ratio of liquid supply flowrate to aerosol sample flowrate; resulting in a measure of oxidative potential in equivalent concentration of BPEAnit-Me per volume of air in nmol.m$^{-3}$ using *Equation 1*. The limit of detection was calculated as three times the standard deviation of a set of 20 s blanks collected while sampling nitrogen.
*Equation 1*

$$C_{ROS} = FR * CF * \frac{q_{ls}}{q_A}$$

Where: $C_{ROS}$ is the concentration of ROS in nmol.m$^{-3}$; FR is the fluorescence response of the spectrometer in unitless counts; CF is the calibration factor calculated from the calibration curve calculated as nmol.L$^{-1}$.counts$^{-1}$; $q_{ls}$ is the liquid sample flowrate in L.min$^{-1}$; and $q_A$ is the aerosol sample flowrate in m$^3$.min$^{-1}$.

## 4.    Results and Discussion

### 4.1 IAC Characterization

#### 4.1.1 Particle Mass Collection Efficiency

The IAC investigation was initially focused on mass collection efficiency in order to directly measure the fraction of aerosol collected into the sample liquid. As particle mass is exponentially proportional to particle size it was necessary to consider
aerosols in terms of mass distribution when evaluating results. To this end, the averaged SMPS particle size distribution of the




ammonium sulphate aerosol sample was fitted with a log-normal distribution and a corresponding mass distribution calculated. These distributions along with those of the ultrafine samples discussed below can be found in the supplementary material. The ultrafine particles ($PM_{0.1}$) correspond to approximately 3 % of the total mass of the sample, whilst the entire mass distribution was in the fine particle size range ($PM_{2.5}$), hence the collection efficiency calculated using this aerosol is referred to fine particle collection efficiency. The averaged collection efficiency and standard error for the collected efficiency for fine particles is calculated as:

$$CE_{fine} = 1.00 \pm 0.04$$

As the majority of the mass in the previous tests was contained in the particles larger than 100 nm (~97 %) this result does not necessarily indicate a high collection efficiency of ultrafine particles. Therefore, these experiments were repeated with the larger ammonium sulphate particles removed from the sample using an impactor as detailed in Sect. 0. The averaged SMPS particle size distribution over the sample collection period was fitted with a log-normal distribution multiplied by a logistic function to emulate the size cut-off generated by the impactor, and a corresponding mass distribution was calculated. The ultrafine particles (<100nm) correspond to approximately 80% of the total mass of the sample, which is considerably higher than the previous size distribution.

The generated particle mass concentrations with the impactor in front of the atomiser (ultrafine particle experiment) were over 60 times lower than the $PM_{2.5}$ experiments. In order to ensure the collected samples contained measurable concentrations of sulphate ions, the flow through the vortex collector was reduced from 1.00 ml.min⁻¹ to 0.15 ml.min⁻¹. This increased the residence time of the sample liquid in the vortex collector for the same aerosol flowrate, resulting in a more concentrated sample.

The averaged collection efficiency and corresponding standard error are calculated as:

$$CE_{ultrafine} = 1.05 \pm 0.06$$

The error of the ultrafine collection efficiency is larger than that of the fine collection due to the significantly lower aerosol sample concentration. However, the results are within error of each other, indicating that the IAC collects ultrafine particles as well as fine particles with a very high efficiency. This is an expected result for steam collection devices, which will typically collect all particle sizes with equally high efficiency due to condensational growth of particles in the growth chamber well into the ultrafine size range.

### 4.1.2 Steam Dilution Factor

The sample liquid entering into the IAC was doped with a known quantity of sodium chloride. The steam dilution factor (SDF) was determined from the change in the Cl⁻ concentrations before and after the liquid feed passed through the vortex collector. Relevant measurements and individual calculated steam dilution factors for each sample can be found in the supplementary material. The average steam dilution factor for a standard liquid supply of 1 mL.min⁻¹ ($SDF_{SF}$) was calculated as:

$$SDF_{SF} = 0.882 \pm 0.004$$





This corresponds to a $0.134 \pm 0.007$ mL.min$^{-1}$ contribution of condensed water into the sample flow.

The averaged steam dilution factor for a lowered liquid supply flowrate of 0.15 mL.min$^{-1}$ ($SDF_{LF}$) was calculated as:

$$SDF_{LF} = 0.527 \pm 0.007$$

$SDF_{LF}$ is lower than that that of the $SDF_{SF}$. This reduction indicates that the ratio of condensed water to sample flowrate has

increased in the liquid sample exiting the PINQ. Despite this, the effective contribution of condensed water to the sample flow

for $SDF_{LF}$ was calculated as $0.136 \pm 0.007$ mL.min$^{-1}$; which is within error of the condensation water contribution calculated

for $SDF_{SF}$. This shows that the condensation rate in the sample is independent of: liquid sample flowrate; PM mass; and particle

number concentration (PNC). This indicates the principle source of condensation in the sample is from inside the cyclone

itself, which is directly proportional to the water flowrate input into the steam generator. This ensures that the steam dilution

factor can be accurately corrected for in all measurements.

Lowering the liquid supply flowrate allows for higher concentrations of aerosol in the sample stream. In theory this could be

further applied to improving the sensitivity of the PINQ. However, the corresponding increasing influence of steam dilution

presents a limitation in the potential application of this. It is essential to keep the contribution of condensed water below 30%

of the total sample (SDF > 0.7) when using a DMSO solution of the BPEAnit probe as larger percentages of water in DMSO

will cause a nonlinear change in the fluorescent signal.

### 4.1.3 Size Dependent Insoluble Particle Collection Efficiency

Ammonium sulphate was selected as the test aerosol for the mass collection efficiency investigation due to its approximately

spherical particle size and detectability with ion chromatography. However, due to its high hygroscopicity it will undergo

condensational growth at a much lower supersaturation than those required for hydrophobic aerosols (Popovicheva et al.,

2008). To ensure that the collection efficiency is independent on the chemical composition of particles, measurements of the

number concentration collection efficiency with hydrophobic insoluble particles were conducted.

Size dependent hydrophobic particle number collection efficiency was measured through the size preclassification of DEHS

aerosol, and corresponding measurements of PNC entering and exiting the PINQ. Collection efficiency was corrected for inlet

and chamber losses which were estimated to be 1 minus the lower limit of the ultrafine mass collection efficiency. The

calculated particle number collection efficiencies at each tested size are shown in Figure 3.



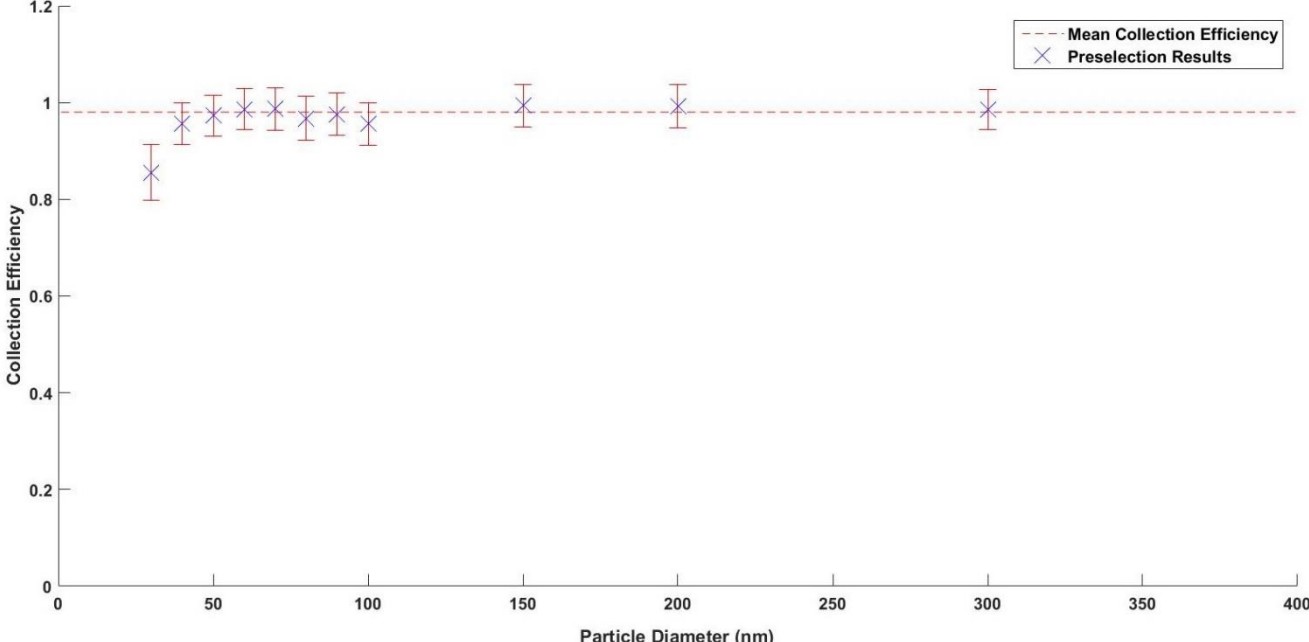

*Figure 3 Calculated collection efficiencies for DEHS preselection measurements with standard error bars and mean collection efficiency*

The mean collection efficiency was calculated as:

$$CE_{PNC} => 0.97$$

As it was not possible to generate sufficient DEHS test particles smaller than 30nm, the cut-off size corresponding to 50% of the maximum collection efficiency can only be extrapolated from the above graph to be < 20 nm. This is a significant improvement over a the earlier wetted-wall cyclone design which reported a collection efficiency of > 0.88 with a cut-off size of 30 nm (Orsini et al., 2008). This ensures that the IAC will be able to capture the majority of particles generated by various atmospheric sources.

## 4.2 PINQ Characterization – Response Time and LOD

The response time of the PINQ was separated into two parameters: a time delay between a source change and the corresponding start of change in fluorescence; and the mixing time taken for the fluorescence response to become completely independent of the previous source. An example of the fluorescence response of the BPEAnit probe when the instrument was switched from the ROS source (cigarette smoke) to nitrogen along with relevant key times is shown in Figure 4. In the 8 samples taken the signal strength of the cigarette smoke was approximately 60 times the magnitude of the background noise.





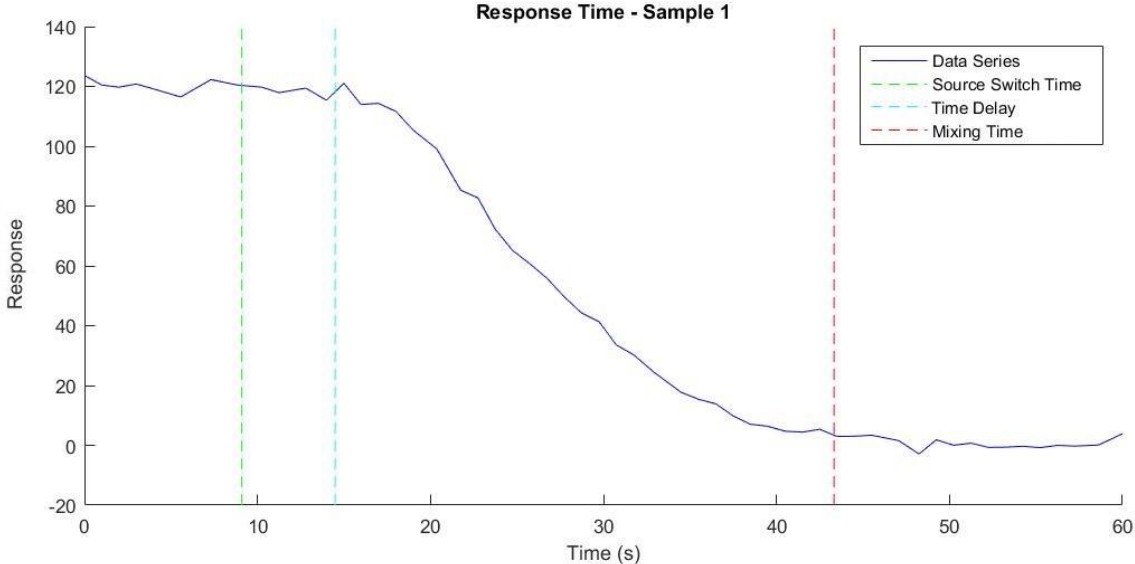

*Figure 4 the fluorescence data series over time for Response Time - Sample 1, with: the time the source was switched from cigarette smoke to nitrogen (Source Switch Time); the time at which the fluorimeter began to measure a decrease in signal (Response Time Delay); and the time at which the signal reduced to background (Mixing Time Delay)*

The averaged values of time delay and mixing time of the PINQ for the eight samples taken and corresponding standard errors were calculated as:

$$time \ delay = 6.3 \ \pm 0.6 \ s$$

$$mixing \ time = 32 \ \pm 1 \ s$$

The time delay of 6.3 s represents a small time correction factor which does not strongly influence instrument performance. In
contrast, the mixing time of 32 s is a key value in determining the limits of instrument sample resolution. This value indicates the minimum averaging time which provides independent data points, and is directly proportional to both: the liquid sample flowrate; and the internal liquid volume of the liquid sample flow path inside the instrument. An increased liquid sample flowrate above the tested 1 mL.min⁻¹ will result in a faster time resolution. However, this will do so at the cost of a lower ratio of collection liquid flowrate to aerosol sample flowrate, diluting the aerosol and reducing instrument sensitivity. Internal liquid
volume was minimized through the use of small bore 1/16" OD tubing for all flowpaths and a custom-made low internal volume fluorescence cell. An unavoidable limiting factor in reducing the mixing time is the internal volume of the cyclone, which cannot be minimized further without impacting instrument performance.

The results of this experiment indicate that the system requires a total of ~40 seconds to reach equilibrium response when connected to an aerosol source. When performing measurements on the oxidative capacity of PM it is necessary to account for
any contributions from the gas phase by alternating sampling between the total aerosol source, and the source after filtering to remove the particle-bound ROS. Therefore, the time resolution of the PINQ is determined by the sum of the total response time, and the time required to average the signal for an accurate measurement. An averaging period of 20 seconds was selected,



leading the PINQ to have a time resolution of 1 minute when performing alternating gas and total phase measurements. A total of 234 blanks of 20 seconds averaging time sampled with nitrogen were collected and normalized as equivalent nmol of BPEAnit-Me per cubic meter of air as detailed in Sect. 0. The limit of detection of the PINQ was determined as three times the standard deviation of these blanks.

$$LOD = 0.08 \ \frac{nmol}{m^3}$$

The PINQ has a lower LOD than other online systems with responses normalized per volume of air sampled for two primary reasons. First, the IAC has a very small ratio of liquid sample flow to aeprosol sample flow. The ensures a more highly concentrated sample than those of particle collector-based systems, which reported LODs between 4 nmol.m$^{-3}$ (Wragg et al., 2016) and 2 nmol.m$^{-3}$ (Zhou et al., 2017). The GAC-ROS system has similar aerosol and liquid flows, resulting in similarly concentrated sample and hence a closer LOD of 0.12 nmol.m$^{-3}$ (Huang et al., 2016a). The second factor contributing to the lower PINQ LOD is the different ROS probe used. The BPEAnit probe does not autooxidise in the same manner that the DCFH probe used by the other system discussed here, resulting in more stable blanks and hence a lower LOD.

## 5. Summary

This manuscript's first focus was on the design and characterization of the IAC as a high efficiency aerosol collector for use in ROS measurements. It is defined as a steam collection device, in which the sample aerosol is continuously mixed with a stream of water vapour to generate a supersaturated mixture, growing the particles into large liquid droplets to ensure high efficiency capture independent of initial particle size. The grown droplets are collected into a continuous liquid sample stream inside a specially designed vortex collector. This component is similar to a miniature PM cyclone, in which a standing liquid vortex is generated in the cone section to collect aerosol directly into liquid. This was designed over a simpler impaction system as it allows the capture of insoluble particles by removing the necessity for an impaction surface; which insoluble particles will typically adhere to rather than collecting into the liquid sample. The vortex collector was designed to be solvent resistant and allow for visual confirmation of the liquid vortex to ensure suitability for application in the PINQ system.

The IAC mass collection efficiency was determined to be within error of 1.00 for both fine and ultrafine particle mass distributions. This result was expected as the condensational growth mechanism used ensures high collection efficiency provided the particles form liquid droplets; and the hydrophilic nature of the ammonium sulphate particles allows for the easy formation of droplets in the supersaturated aerosol regardless of initial particle size. As the IAC must collect particle independent of chemical composition for use in the PINQ system, the number collection efficiency of highly hydrophobic DEHS test particles was also investigated. Using these particles the number collection efficiency was found to be > 0.97 with a cut-off size of < 20 nm. This result shows that the IAC is capable of collecting particles with a high efficiency independent of particle size and composition.



The PINQ was developed to measure oxidative potential using the BPEAnit chemical probe in conjunction with the IAC. A 1 µM solution of BPEAnit in DMSO is used as the sample collection liquid, with particles collected directly into the probe solution inside the IAC vortex collector. The rapid mixing of liquid inside the vortex coupled with the diffusion limited reaction between the probe and any ROS collected ensures the liquid exiting the IAC is fully reacted. The sample liquid is then

debubbled and input into a specially designed flow-through through fluorimeter. Finally, the fluorescence response measured is converted into oxidative potential through calibrations performed on known concentrations of BPEAnit-ME, and are expressed in nmol.m$^{-3}$.

In order to quantify the particle phase ROS signal, it is necessary to alternately sample HEPA filtered and unfiltered air due the potential of contributions from the gas phase. This method was chosen over the use of a gas denuder as the gas phase can

contain significant concentrations of ROS relevant to oxidative capacity (Stevanovic et al., 2017). The quantification of this gas phase signal is a complex issue as it will likely be dependent upon each gas species solubility in both water and DMSO. For this reason the quantification of the gas phase collection efficiency of the PINQ is beyond the scope of this paper, and any gas phase data presented in manuscripts will be semi-quantitative in nature until further study is undertaken.

Experiments on response time with a standard sample flowrate of 1 mL/min indicate that after switching sources the PINQ

signal takes ~ 40 seconds to stabilize. Therefore, with a 20 second sample averaging time the sample time resolution of the instrument is one minute when alternating between HEPA-filtered and unfiltered samples. With this time resolution the LOD of the instrument was determined to be 0.08 nmol.m$^3$. Both the time resolution and LOD of the instrument are considerably lower than other instruments currently found in literature, indicating the PINQ is a viable candidate for the quantification of aerosol oxidative potential.

### Data availability

Collection efficiency of PINQ data is available as part of an online Supplement.





**Author contribution**

R. A. B Improved and finalised the design of PINQ; Made experimental set-up, performed data analysis and wrote the manuscript

S.S Made initial design of the device; contributed to the experimental design, assisted with data interpretation; reviewed the manuscript

S.B Reviewed the manuscript; contributed to the experimental design and data interpretation.

Z. D. R Came up with the original idea; contributed to the experimental design, assisted with data interpretation; reviewed the manuscript

**Competing interests**

The authors declare that they have no conflict of interest

**Acknowledgements**.

The authors wish to thank the following groups/institutions for their contribution to this work: the aerosol physics group at

Tampere University of Technology (TUT) for their assistance in the characterisation of the initial version of the device; Central

Analytical Research Facility (CARF) for their assistance with ion chromatography analysis, and the Design and Manufacturing

Centre (DMC) team from Queensland University of Technology for their help with the design and building of this device. This

research was funded by: ARC Discovery grant DP120100126 Fundamental study into the role of the organic fraction on the

toxicity of combustion generated airborne particles and EU FP7 Project ID: 308524, CITI-SENSE Development of sensor-

based Citizens' Observatory Community for improving quality of life in cities.



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
