# Peer review of "An instrument for the rapid quantification of PM-bound ROS: the Particle Into Nitroxide Quencher (PINQ)"

_Atmospheric Measurement Techniques, 2018_

## Referee Comment (RC1) · Anonymous Referee #2 · 24 Nov 2018

This paper reports on the development of a steam-particle-growth system coupled to a cyclone for droplet collection followed by analysis of the liquid for particle bound ROS with a fluorescence probe. Recently, a number of papers have been published describing various designs aimed at measuring particle-bound ROS. This paper adds an additional method to this group. The authors have done a series of careful experiments testing the performance of the instrument, however there are a number of major issues to address. 1) The particle collection system of the instrument is nearly identical to other instruments already reported in the literature, with the exception that the cyclone design differs; exactly how it differs is not totally clear since exact details are not provided. 2) The instrument described measures only ROS associated with particles, yet

it equates measurements of particle bound ROS with other assays that measure completely different particle species associated with the aerosols ability to induce oxidative stress. This conflation adds to the confusion that seems to exist in the new field of aerosol research. Prior to publication, these two major issues should be addressed. Details are provided below, along with other minor issues to consider.

Major Issues

A major issue is that the instrument described in this paper is essentially identical to a PILS with a cyclone droplet collector instead of impactor. Orsini et al (2008) describes an instrument with PILS coupled to a cyclone (which the authors cite), and Peltier et al (2007) specifically describes and tests a PILS-mini cyclone system (not cited in this work). Since the proposed instrument is so similar to these existing instruments, details of how the new instrument differs, such as what are the technical advances in this new instrument over existing technology. For example, why not just utilize the existing methods? It appears that the authors are implying that the cyclone design is novel, ie, that it produces what they call a standing vortex. If this is indeed the main novel feature, the exact details of how the cyclone was designed and constructed to achieve this should be discussed in more detail. As it stands it is doubtful a reader could reproduce the results of this paper due to insufficient detail.

A second major issue with this paper is the conflation of particle bound ROS and aerosol species that produce ROS in vivo, which is now often referred to as oxidative potential (OP). The title states that this is a method to measure PM OP, but it is more precisely a measure of ROS associated with particles. The title should be more specific, eg, possibly changed to something like. . . rapid quantification of ROS associated with particles... Also, in the abstract it should be clearly stated that particle bound ROS is being measured. Throughout the paper care should be taken to delineate the two. Thus, it would be best not to equate what is being measured in this work with OP, eg, Pg 2 line 17 states: In order to achieve ROS quantification, termed herein oxidative potential, ...

The two, particle ROS and OP should be delineated as they are associated with different aerosol chemical species and have potentially different health effects. Particle ROS, what the authors call exogenous ROS is what this paper is measuring and which has not been associated with any adverse health effects in population studies; at least this reviewer does not know of any. Maybe the authors can add citations supporting why particle-bound ROS is an important thing to measure. An example may be direct inhalation of combustion products, like cigarette smoke? In contrast to particle-bound ROS, endogenous ROS, which is often referred to as OP and typically measured with the DTT or GSH assay, has been associated with adverse cardiorespiratory adverse health effects in some studies (Abrams et al 2018; Bates et al., 2015; Weichenthal et al., 2016; Yang et al., 2016). Thus, it is strongly suggested that the term oxidative potential be removed throughout the paper and replaced with particle-bound ROS, or similar notation, except in the cases where DTT assay is specifically referenced, eg, Table 1. Overall, the point is that this research area lacks an agreed upon terminology, but no matter what the authors decide to call what they measure, it is very important that it be made clear that it is fundamentally different than certain other assays that measure aerosol chemical species that generate ROS in vivo (eg, DTT or GSH, etc).

Related to this, if particle-bound ROS is so reactive and has a short life-time, the justification for this instrument, why would one expect it to be a significant health hazard to a large segment of the population? It seems the instrument is most useful for measuring particle-bound ROS associated with very fresh combustion emissions. Where specifically would one then expect to deploy this instrument. A discussion along these lines should be added. This would further help clarify the difference between what this instrument is measuring vs methods using the DTT or GSH assays (ie, methods measuring aerosol oxidative potential not particle bound ROS).

Minor Issues

Pg 2 last line is not correct as two online DTT systems have been developed, see Puthussery et al. (2018), and Eiguren-Fernandez, et al, (o-MOCA), which is cited.

Last paragraph of section 1.3.4: There is a difference between a solid insoluble particle and a hydrophobic particle. Can the authors give an example of an insoluble particle bound ROS species? For oxidative potential, this is extensively discussed in Fang et al, (2017) This is an important question since the authors are using this as design criteria. (More on this below).

Section 2.2.4, What specific system used a copper steam generation system the authors refer to? This is not common practice in most steam systems, including the commercially available PILS. References to the copper system should be removed unless specific instruments using it can be identified.

In the particle mass collection efficiency method was the aerosol neutralized after nebulization for the IAC leg, as done for the SMPS leg? If not the reason for not doing this and implications should be discussed since one may expect highly charged particles. Also, how is the impactor affected by these highly charged particles (if there was no neutralization)?

Why is the impactor installed before the particles are dried? Was the cut size of the impactor actually 0.1 um or did it remove larger droplets, but ended up effectively removing dried particles with diameters less than 0.1 um?

Pg 16, line 8, if the particles are dried, depending on the RH achieved, the ammonium sulfate may not be spherical. Why not do a sensitivity test to see how the findings change if say the DMA sizing is corrected assuming non-spherical particles.

Pg 16, line 10, why does the the Dp log scale make the area under the curve not proportional to fraction of overall mass when the size distribution is plotted as dN/dlogDp? That is precisely the point of the size distribution function.

Pg 17 line 4, typo, 0?

Section 4.1.3. What is the difference between a hydrophobic particle and an insoluble particle? Is DEHS insoluble in very dilute systems? Does DEHS remain as the

original sizes generated in the droplet collection system (ie, cyclone)? The point is the one thing that is unique about this instrument is the claim that it can measure at near 100% efficiency insoluble particles, but only one form is tested. What about collection efficiency of solid particles? Will the instrument actually collect solid particles and thus have the ability to measure ROS associated with solid particle surfaces, say for example, fresh soot particles. Are comparisons of this vortex cyclone to the Orsini et al or Peltier et al mini-cyclone valid since their test were done with truly solid particles (PSL or soot)?

References

Abrams, J., R. J. Weber, M. Klein, S. E. Samat, H. H. Chang, M. J. Strickland, V. Verma, T. Fang, J. T. Bates, J. A. Mulholland, A. G. Russell, and P. E. Tolbert (2017), Associations between ambient fine particulate oxidative potential and cardiorespiratory emergency department visits, Envir. Health Perspectives, 25(10), 1-9.

Bates, J. T., R. J. Weber, J. Abrams, V. Verma, T. Fang, M. Klein, M. J. Strckland, S. Sarnat, H. Chang, J. A. Mulholland, P. E. Tolbert, and A. G. Russell (2015), Reactive Oxygen Species in Atmospheric Particulate Matter Suggest a Link to Cardiorespiratory Effects, Envir. Sci. Technol, 49, 13605-13612

Fang, T., L. Zeng, D. Gao, V. Verma, A. Stefaniak, and R. J. Weber (2017), Ambient Size Distributions and Lung Deposition of Aerosol Oxidative Potential: A Contrast Between Soluble and Insoluble Particles Envir. Sci. Technol, 51, 6802-6811.

Peltier, R. E., R. J. Weber, and A. P. Sullivan (2007), Investigating a liquid-based method for online organic carbon detection in atmospheric particles, Aerosol Sci. Tech., 41, 1117-1127

Puthussery, J. V., C. Zhang, and V. Verma (2018), Development and field testing of an online instrument for measuring the real-time oxidative potential of ambient particulate matter based on dithiothreitol assay, Atmos. Meas. Tech., 11, 5767-5780.

Weichenthal, S. A., E. Lavigne, G. J. Evans, K. J. G. Pollitt, and R. T. Burnett (2016), PM2.5 and Emergency Room Visits for Respiratory Illness: Effect Modification by Oxidative Potential, Am J Resp Crit Care Med, 194, 577-586.

Yang, A., N. A. H. Janssen, B. Brunekreef, F. R. Cassee, G. Hoek, and U. Gehring (2016), Children's respiratory health and oxidative potential of PM2.5: the PIAMA birth cohort study, Occup. Environ. Med, 73, 154-160.

---

## Referee Comment (RC2) · Anonymous Referee #1 · 27 Nov 2018

The authors present an instrument for quantifying PM associated ROS using 9,10-bis (phenylethynyl) anthracene-nitroxide, (BPEAnit), as probe. Particles were collected directly by a combination of a steam-aided growth chamber and a miniature cyclone with added reaction solution. Particle losses, sensitivity and response time are discussed in detail, exemplarily by analyzing cigarette smoke.

Major comments:

1. The instrument is not sufficiently described, e.g. size data of the "Insoluble Aerosol Collector" (IAC) should be added to estimate the flow regime.

2. The authors stated that the instrument is measuring ROS of particle matter (PM) – is

that correct? ROS in gasphase (not associated completely with PM, e.g. H2O2) should be collected too, if the gas/particle distribution is shifted towards particles (droplets) after adding condensing water vapor. The relevance of this process should be discussed. If important, loss of gaseous ROS must be determined or measures as e.g. installing a charcoal denuder prior to PM collection should be considered. Perhaps the term "particle" should be replaced by "aerosol", defined as mixture of gas, particles and droplets.

3. In section 1 several probes and collection devices reviewed. It will be helpful, if the probe dependent targeted ROS species (O-centered-, C-centered-radicals, . . .) will be discussed more detailed to enable interpretation of the different end-points.

Minor comments:

Page1 line 18: Replace ". . .(Nel, 2005)(Penttinen et al.,2001); and . . ." by ". . .(Nel, 2005; Penttinen et al.,2001) and . . ."

Page 2, line 26: ". . .are only sensitive to narrow ranges of ROS species; making them unsuited for quantification of total oxidative potential." Here a more detailed discussion on determined ROS species will be helpful

Section 1.3: Subtitle should be replaced by "online collection techniques", as chemical assays are discussed in section 1.2.

Page 5 line 19 and page 6 line 9 and 10 and some other occurences: . . ."sample resolution" means ". . .time resolution. . ."

Page 7 line 7-8: Where is the aerosol flow rate measured?

Page 8 line 9: A mass flow in g.min-1 will be more exact, additionally the dew point temperature of the supersaturated aerosol flow will be a vivid parameter.

Page 8 line 13: Why is the excess condensation drain volume not measured for calculation of the steam dilution factor? Can the excess drain water be analysed further for

cross checking particle losses and removal of gaseous components?

Section 3.1.2: Did the authors check, if the charcoal denuder really removes all ethanol vapour from DEHS particles? If there is only a small amount of ethanol left on the surface of the particles, the still have a hydrophilic character.

Page 11 line 29-30: Particle mass increases with 3rd order to particle diameter/size, not exponentially Page 12 line 10 and page 16 line 3: "Sect. 0" is not defined

---

## Author Comment (AC1) · 30 Dec 2018

Comment 1

The instrument is not sufficiently described, e.g. size data of the "Insoluble Aerosol Collector" (IAC) should be added to estimate the flow regime.

Response 1

The authors agree, and have provided information on the internal IAC dimensions in Sect 2.2. in the updated manuscript.

Comment 2

[Figure]

The authors stated that the instrument is measuring ROS of particle matter (PM) – is that correct? ROS in gas phase (not associated completely with PM, e.g. H2O2) should be collected too, if the gas/particle distribution is shifted towards particles (droplets) after adding condensing water vapor. The relevance of this process should be discussed. If important, loss of gaseous ROS must be determined or measures as e.g. installing a charcoal denuder prior to PM collection should be considered. Perhaps the term "particle" should be replaced by "aerosol", defined as mixture of gas, particles and droplets.

Response 2

The collection of gas phase species was noted in Sect 5. However it has been moved and expanded upon in a dedicated Sect 4.3. in the updated manuscript. During measurement the instrument should alternate between filtered and unfiltered samples in order to account for the gas phase contribution. Another approach would be using a denuder as described by the reviewer. However, the concern with this method would be the losses of ultrafine particles in the denuder prior to sampling (Stevanovic et al., 2015). Currently the IAC is only characterized for PM collection, with the gas phase collection of interest but still in the very early stages of investigation due to its inherent difficulty. The authors feel it would be premature to claim aerosol collection until the gas phase is fully characterized. Instead, in any sampling campaign the inlet will be periodically switched to sample after a filter to account for gas phase measurements to present a purely particle phase result. Gas phase data can be interpreted to some degree, however until characterization it is semi-quantitative at best. Accounting for the gas phase contribution was not performed in the response time experiments as the total aerosol response was of interest.

Comment 3

In section 1 several probes and collection devices reviewed. It will be helpful, if the probe dependent targeted ROS species (O-centered-, C-centered-radicals, . . .) will

be discussed more detailed to enable interpretation of the different end-points.

Response 3

Some information on the sensitivity of the different major probes discussed has been added in the updated manuscript in Sect 1.2.

Comment 4

Page1 line 18: Replace ". . .(Nel, 2005)(Penttinen et al.,2001); and . . ." by ". . .(Nel, 2005; Penttinen et al.,2001) and . . ."

Response 4 This has been corrected in the updated manuscript in Page 1, Line 18-19.

Comment 5

Page 2, line 26: ". . .are only sensitive to narrow ranges of ROS species; making them unsuited for quantification of total oxidative potential." Here a more detailed discussion on determined ROS species will be helpful

Response 5

The discussion regarding difference ROS species and the use of phrase oxidative potential has been added in Sect 1.2.

Comment 6

Section 1.3: Subtitle should be replaced by "online collection techniques", as chemical assays are discussed in section 1.2.

Response 6

This has been corrected in the updated manuscript at Page 3, Line 29.

Comment 7

Page 5 line 19 and page 6 line 9 and 10 and some other occurences: . . ."sample resolution" means ". . .time resolution. . ."

Response 7

All incidents of sample resolution have been replaced with time resolution in the updated manuscript.

Comment 8

Page 7 line 7-8: Where is the aerosol flow rate measured?

Response 8

The aerosol flowrate is set prior to, and checked after measurement using a TSI 4043 mass flow meter placed in line between the sampling line and the PINQ inlet. This was found to be stable over long periods (24 hours with no significant variance) and more reliable than a mass flow controller which suffered from issues with condensation. This has been updated in the current manuscript at Page 7, Line 16-17.

Comment 9

Page 8 line 9: A mass flow in g.min-1 will be more exact, additionally the dew point temperature of the supersaturated aerosol flow will be a vivid parameter.

Response 9

A mass flow of g.min-1 would be more exact, however the peristaltic pump is a volumetric device and hence a volume flow is provided. 1.5mL.min-1 provides an excess of supersaturation which will still be achieved with small variances caused by density changes caused by changes in different laboratory environments. If the instrument is operated in extreme environments with significantly elevated or lowered temperatures tests would have to be performed to ensure viability. This clarification has been included in the updated manuscript. In regards to dew point temperature, the goal of the system is to provide excessive supersaturation to grow even the smallest hydrophobic particles into liquid droplets. As such the resultant aerosol will be well below the dew point temperature in all circumstances, and hence it is not a parameter of interest here.

Comment 10

Page 8 line 13: Why is the excess condensation drain volume not measured for calculation of the steam dilution factor? Can the excess drain water be analysed further for cross checking particle losses and removal of gaseous components?

Response 10

The steam dilution factor cannot be calculated using the chamber drain line, as you would also need to account for the water collected in the liquid trap and any condensation in tubing connected to the exhaust line of the instrument. The IC analysis during the ammonium sulphate collection experiments provided a significantly more accurate and direct way of measuring this value at Page 8 Line 1-4. As shown in the manuscript, the PM mass collected by the IAC is within error of the aerosol sample mass. Whilst there is inevitably some particle losses in the chamber, these losses constitute a very small mass portion of the aerosol. Beyond issues regarding how effectively the chamber is washed by the condensation to remove particle deposition, the concentrations of ammonium sulphate in the mass concentration in the chamber drain would be below the limit of detection of the IC system. The removal of gaseous components could be potentially significant and is of potential interest in the future. However, the gas phase collection is not considered in this publication.

Comment 11

Section 3.1.2: Did the authors check, if the charcoal denuder really removes all ethanol vapour from DEHS particles? If there is only a small amount of ethanol left on the surface of the particles, the still have a hydrophilic character.

Response 11

The charcoal denuder will remove a substantial portion of the ethanol from the aerosol source. The aerosol is then effectively diluted at a ratio of 10 to 1 inside the SMPS due to the sheath air interaction, and diluted again at a ratio of 15.7 to 1 when mixed with the

makeup aerosol to generate the required flowrate. Given the volatility of ethanol is so high and the combination of two dilution stages the authors are confident that ethanol will only be present in trace concentrations and will not influence the hydrophobicity of the particles.

Comment 12

Page 11 line 29-30: Particle mass increases with 3rd order to particle diameter/size, not exponentially

Response 12

This has been corrected in the updated manuscript at Page 12, Line 16.

Comment 13

Page 12 line 10 and page 16 line 3: "Sect. 0" is not defined

Response 13

This has been corrected in the updated manuscript.

Please also note the supplement to this comment:
https://www.atmos-meas-tech-discuss.net/amt-2018-333/amt-2018-333-AC1-supplement.pdf
* * *
[Figure]

**Supplement:**

**Supplementary Material (SM) For:**

[revised manuscript text omitted]

Whilst ammonium sulphate tends towards sphericality at small sizes, a dynamic shape factor of 1.02 can be used to correct for a minor aspherical variations (Biskos et al., 2006). In this circumstance applying this factor will negligibly alter the calculations of the percentage of mass in the ultrafine region, as the error margins of the fitted curves are significantly higher than this effect. Hence, to avoid needlessly complicating the analysis the aerosol here is considered to be approximately spherical.

As the collection efficiency calculated using the results of this section are for mass collection, it is important to consider the particles in terms of their mass rather than number distributions. To achieve this, the particle size distributions were fitted with log normal curves of the form given in Equation S5 using the Matlab curve fitting toolbox.

*Equation S5*

$$N = ae^{\left(\frac{\ln(D)-\ln(b)}{c}\right)^2}$$

Where: N is the number concentration; D is particle diameter; and a, b and c are the fitted coefficients.

To convert the fitted number distributions to mass distributions, the particles were assumed to be spherical and homogenous. This resulted in the mass distribution curve of the form shown in Equation S6.

*Equation S6*

$$M = ae^{\left(\frac{\ln(D)-\ln(b)}{c}\right)^2} * \frac{4}{3}\pi\left(\frac{D}{2}\right)^3 * \rho$$

Where: M is particle mass; and $\rho$ is particle density.

The averaged SMPS particle size distribution of the ammonium sulphate $PM_{2.5}$ aerosol sample was fitted with a log-normal distribution and a corresponding mass distribution calculated as shown in *Figure S3*. The ultrafine particles ($PM_{0.1}$) correspond to approximately 3 % of the total mass of the sample, whilst the entire mass distribution was in the fine particle size range ($PM_{2.5}$).

[Figure]

*Figure S3 a) is a graph of the averaged SMPS data for the fine particle concentration measurements and a fitted log-normal distribution with 95 % confidence intervals and an $R^2$ value of 0.9769. b) shows the calculated mass distribution curve with 95 % confidence intervals. The ultrafine mass portion is shaded, corresponding to ~3% of the total mass.*

When an impactor was used to remove larger particles the size cut-off was modelled using a logistic function. Therefore, the fitted number distribution took the form of Equation S7.

*Equation S7*

$$N = ae^{\left(\frac{\ln(D)-\ln(b)}{c}\right)^2} * \frac{1}{1 + e^{d(x-f)}}$$

5     Where: d and f are fitted coefficients.

As with the previous distributions, to convert the fitted number distributions to mass distributions the particles were assumed to be spherical and homogenous. This resulted in the mass distribution curve of the form shown in Equation S8.

*Equation S8*

$$M = ae^{\left(\frac{\ln(D)-\ln(b)}{c}\right)^2} * \frac{1}{1 + e^{d(x-f)}} * \frac{4}{3}\pi \left(\frac{D}{2}\right)^3 * \rho$$

[revised manuscript text omitted]

---

## Author Comment (AC2) · 30 Dec 2018

Reviewer Opening Statement

This paper reports on the development of a steam-particle-growth system coupled to a cyclone for droplet collection followed by analysis of the liquid for particle bound ROS with a fluorescence probe. Recently, a number of papers have been published describing various designs aimed at measuring particle-bound ROS. This paper adds an additional method to this group. The authors have done a series of careful experiments testing the performance of the instrument, however there are a number of major issues to address. 1) The particle collection system of the instrument is nearly identical to

other instruments already reported in the literature, with the exception that the cyclone design differs; exactly how it differs is not totally clear since exact details are not provided. 2) The instrument described measures only ROS associated with particles, yet it equates measurements of particle bound ROS with other assays that measure completely different particle species associated with the aerosols ability to induce oxidative stress. This conflation adds to the confusion that seems to exist in the new field of aerosol research. Prior to publication, these two major issues should be addressed.

Author Response Statement

The lines listed by the reviewer in their comments corresponds to the original submitted manuscript and not the one published for online discussion. As such the lines and pages listed by the reviewer or not relevant to the comments.

Comment 1

A major issue is that the instrument described in this paper is essentially identical to a PILS with a cyclone droplet collector instead of impactor. Orsini et al (2008) describes an instrument with PILS coupled to a cyclone (which the authors cite), and Peltier et al (2007) specifically describes and tests a PILS-mini cyclone system (not cited in this work). Since the proposed instrument is so similar to these existing instruments, details of how the new instrument differs, such as what are the technical advances in this new instrument over existing technology. For example, why not just utilize the existing methods? It appears that the authors are implying that the cyclone design is novel, ie, that it produces what they call a standing vortex. If this is indeed the main novel feature, the exact details of how the cyclone was designed and constructed to achieve this should be discussed in more detail. As it stands it is doubtful a reader could reproduce the results of this paper due to insufficient detail.

Answer 1

The authors disagree with this statement strongly. This paper is titled "An instrument for

the rapid quantification of PM oxidative potential: the Particle Into Nitroxide Quencher (PINQ)" and as such is a paper regarding an instrument (the PINQ) which measures the oxidative potential of PM. The PILS is a particle collector, which is an entirely different category of instrument altogether. The PINQ uses a combination of chemistry and purpose built collection and measurement stages which is clearly different than any other instrument published or commercially available for this purpose. This has been done in order to create an OP measurement instrument with as high time resolution and sensitivity as possible (< 1 minute). The authors believe the reviewer has mistaken the purpose of this paper as a discussion of the IAC, the particle collection stage of the PINQ. A large portion of the paper is dedicated to the design and collection efficiency of the IAC, which is entirely necessary given that the PINQs performance is predicated on this and it has not been described in a publication previously. However, the IAC is a single, if important piece in the larger PINQ. It is predominantly named separately in order for readers to easily distinguish between discussion regarding the particle collection stage and those regarding the PINQ instrument as a whole. Despite this important note, the IAC itself is distinct from the PILS and other commercially available and published steam collection devices. The initial inspiration for the IAC does indeed come from the modified PILS with a "wetted-wall cyclone" described in Orsini et al (2008). However, the variations from this system are substantial and include (but are not limited to): A liquid cooling system on the growth chamber to improve heat removal efficiency for elevated temperature applications; a completely new steam generator design to improve system stability and flowrate changes; and the development of a solvent resistant version of the wetted-wall cyclone compatible with DMSO which was termed a vortex collector. The decision to introduce this new term is not due to a change in mechanism from the one described by Orsini (2008), in this papers abstract it also describes "a standing liquid vortex which coats the inside deposition surface". Instead this was intended to separate the design clearly from that of a traditional PM cyclone as their purposes and design focus are different as discussed in Sect. 2.2. Some modifications have been made to the manuscript in Sect. 2.2 to emphasize these differences and

their purpose in the development of an oxidative potential monitor. The authors were not aware of Peltier et al (2007). It is relevant to this study and has been integrated into the manuscript. As to the comment regarding the reproducibility of this study, all key conceptual information and design considerations, as well as published resources used in the development, construction and testing of this system are provided. As requested by Referee #1, more detail has been added on internal dimensions of the system. Further information regarding detailed construction information, schematics, etc would require a manuscript several times the length of the current one and would provide little utility. Instead, the following statement has been added to the data availability: For a more detailed description on the design and construction of the instrument or interest in instrument evaluation, please contact the corresponding author.

Comment 2

A second major issue with this paper is the conflation of particle bound ROS and aerosol species that produce ROS in vivo, which is now often referred to as oxidative potential (OP). The title states that this is a method to measure PM OP, but it is more precisely a measure of ROS associated with particles. The title should be more specific, eg, possibly changed to something like . . . rapid quantification of ROS associated with particles ... Also, in the abstract it should be clearly stated that particle bound ROS is being measured. Throughout the paper care should be taken to delineate the two. Thus, it would be best not to equate what is being measured in this work with OP, eg, Pg 2 line 17 states: In order to achieve ROS quantification, termed herein oxidative potential, ... The two, particle ROS and OP should be delineated as they are associated with different aerosol chemical species and have potentially different health effects. Particle ROS, what the authors call exogenous ROS is what this paper is measuring and which has not been associated with any adverse health effects in population studies; at least this reviewer does not know of any. Maybe the authors can add citations supporting why particle-bound ROS is an important thing to measure. An example may be direct inhalation of combustion products, like cigarette

smoke? In contrast to particle-bound ROS, endogenous ROS, which is often referred to as OP and typically measured with the DTT or GSH assay, has been associated with adverse cardiorespiratory adverse health effects in some studies (Abrams et al 2018; Bates et al., 2015; Weichenthal et al., 2016; Yang et al., 2016). Thus, it is strongly suggested that the term oxidative potential be removed throughout the paper and replaced with particle-bound ROS, or similar notation, except in the cases where DTT assay is specifically referenced, eg, Table 1. Overall, the point is that this research area lacks an agreed upon terminology, but no matter what the authors decide to call what they measure, it is very important that it be made clear that it is fundamentally different than certain other assays that measure aerosol chemical species that generate ROS in vivo (eg, DTT or GSH, etc). Related to this, if particle-bound ROS is so reactive and has a short life-time, the justification for this instrument, why would one expect it to be a significant health hazard to a large segment of the population? It seems the instrument is most useful for measuring particle-bound ROS associated with very fresh combustion emissions. Where specifically would one then expect to deploy this instrument. A discussion along these lines should be added. This would further help clarify the difference between what this instrument is measuring vs methods using the DTT or GSH assays (ie, methods measuring aerosol oxidative potential not particle bound ROS).

Answer 2

The research supporting the association between the OP and health has been limited (to very few studies, mainly from 1 research group) and it is still in its pioneer stage. Insufficient evidence in this regard cannot be used to predict and suggest an explanation for observed health effects upon exposure to particulate matter. Oxidative potential that depicts the presence and concentration of in-vivo present (or generated) redox species, measurable by the DTT, AA or GSH, is not a widely accepted terminology to the best of our knowledge. There is no means to detect the total oxidative potential using any cell-free assay and thus it would not be accurate and representative to use the term oxidative potential for any of these techniques, including the DTT.

In addition, ROS present on particles will also contribute to the OP. It has not been established what is the contribution of exogeneous ROS to the total OP, but excluding it will be inaccurate. It is also unclear what is the detection capacity of DTT for different particle types. So, saying that DTT is measuring the OP is also a bold statement as this approach is limited to certain solvents and reactive species. We agree that in the lack of data supporting the link between the ROS and OP, we should use more precise terminology. To this end, a short note on the inability of current probes to described as a measure of oxidative capacity has been added in Sect. 1.2. Furthermore, the BPEAnit measurements performed with the PINQ are now referred to as measurements of PM-bound ROS. Authors cannot agree with few comments highlighted by the Reviewer. "The two, particle ROS and OP should be delineated as they are associated with different aerosol chemical species and have potentially different health effects." This statement is not supported by any piece of literature data and thus we could not consider modifying the discussion in the manuscript. "Maybe the authors can add citations supporting why particle-bound ROS is an important thing to measure". Authors strongly believe that this is more than obvious and that the research in the area in the last 10 years demonstrates this. Importance of particle-bound ROS is beyond cardio-respiratory health effects. We believe that any atmospheric scientist is aware of the importance of particle-bound ROS. Once in the atmosphere, they will be oxidised easier, creating potentially more toxic form of pollutants, secondary pollutants. ROS on particles will change organic composition, that is found to be responsible for the OP, regardless of the approach used to measure it. In regards to the last question: "Related to this, if particle-bound ROS is so reactive and has a short life-time, the justification for this instrument, why would one expect it to be a significant health hazard to a large segment of the population? It seems the instrument is most useful for measuring particle-bound ROS associated with very fresh combustion emissions. Where specifically would one then expect to deploy this instrument." It is well known that the ROS on particles can be short-living and long-living. Persistent free radicals are also very important for the measurement of the oxidative reactivity of particles. This instrument is designed to measure the reactivity of particles in real-time and is not limited to the fresh combustion emissions. It can be deployed for atmospheric measurements, chamber studies, mechanistic studies, just to name a few. It can be also utilised to detect the change in oxidative reactivity over long periods of time. As indicated before, there is no doubt that the particle-bound ROS is very important for observed health effects, but only a very well designed, large cohort study (or a few of them) can shed more light into what is posing a health hazard to the large population.

Comment 3

Pg 2 last line is not correct as two online DTT systems have been developed, see Puthussery et al. (2018), and Eiguren-Fernandez, et al, (o-MOCA), which is cited.

Answer 3

Page 3, Line 3 has been clarified to indicate that the limiting factor is in reference to time resolution of the instruments possible rather than the inability to create a real time system. For the referees interest there is also a third DTT system (Sameenoi et al., 2012). Puthussery et al. (2018) is not included as it was published after the submission of this manuscript.

Comment 4

Last paragraph of section 1.3.4: There is a difference between a solid insoluble particle and a hydrophobic particle. Can the authors give an example of an insoluble particle bound ROS species? For oxidative potential, this is extensively discussed in Fang et al, (2017). This is an important question since the authors are using this as design criteria. (More on this below).

Answer 4

The terms hydrophobic and insoluble were incorrectly used interchangeably as, in the case of the DCFH and DTT assays, the chemistry is performed in an aqueous solution and hence hydrophobic particles tend to be insoluble in that context. This has

been corrected in the updated manuscript in several places including Sect 1.3.4. The emphasis here is not on insoluble ROS, but on insoluble particles. The PINQ system collects these particles directly into the BPEAnit in DMSO solution, removing the requirement for particle solubility in the collection liquid used in several of the systems discussed in Sect 1.3.

Comment 5

Section 2.2.4, What specific system used a copper steam generation system the authors refer to? This is not common practice in most steam systems, including the commercially available PILS. References to the copper system should be removed unless specific instruments using it can be identified.

Answer 5 The authors agree that the use of copper is not common in these applications and have removed reference to it from the manuscript in Sect 2.2.4.

Comment 6

In the particle mass collection efficiency method was the aerosol neutralized after nebulization for the IAC leg, as done for the SMPS leg? If not the reason for not doing this and implications should be discussed since one may expect highly charged particles. Also, how is the impactor affected by these highly charged particles (if there was no neutralization)?

Answer 6

Neutralization in an SMPS is widely accepted standard practice to ensure correct sizing. The entire flow path was conductive to minimize electrostatic losses. Beyond this the authors do not believe that aerosol charge has any influence on the setup and the results presented. Charge plays no role in IAC collection and would not influence the impactor performance.

Comment 7

Why is the impactor installed before the particles are dried? Was the cut size of the impactor actually 0.1 $\mu$m or did it remove larger droplets, but ended up effectively removing dried particles with diameters less than 0.1 $\mu$m?

Answer 7

The purpose of the impactor was not to generate a sharp cut-off at 100nm, but to make the resultant size distribution as small as possible so that the mass distributions were predominantly in the ultrafine range. Placing the impactor first lowered the effective cut-off size once the particles are dried, enhancing this effect. The cut-off size can be estimated as $\sim$74 nm based on the logistic function portion of the fitted curve for the particle size distribution with the impactor present (See Figure S4). This has been corrected and expanded in the updated supplementary material in Page S3, Lines 9-15.

Comment 8

Pg 16, line 8, if the particles are dried, depending on the RH achieved, the ammonium sulfate may not be spherical. Why not do a sensitivity test to see how the findings change if say the DMA sizing is corrected assuming non-spherical particles.

Answer 8

Ammonium sulphate particles are not perfectly spherical, however smaller particles tend towards sphericality (Zelenyuk et al., 2006). The particles in question here have a mean diameter between 30 - 40 nm which were dried to $\sim$20 % RH. Whilst inevitably they will be slightly aspherical, they will only be marginally so. The most suitable dynamic shape factor found would be 1.02 (Biskos et al., 2006), which results in no significant change to the mass percentage present in the ultrafine range due to the significantly higher contribution of uncertainty from the function fitting ($80 \pm 10$). Hence it represents a needless complication of the analysis. The supplementary material has been edited to add this justification of the sphericity assumption in Page S4, Lines

1-5.

Comment 9

Pg 16, line 10, why does the the Dp log scale make the area under the curve not proportional to fraction of overall mass when the size distribution is plotted as dN/dlogDp? That is precisely the point of the size distribution function.

Answer 9

dN/dlogDp normalizes each bin concentration by its size width in order to allow comparison between instruments with different bin resolutions. It is still a measure of particle size, not mass. Particle mass of a chemically homogenous particle is equal to particle volume multiplied by density, and hence is not directly proportional to particle size. As stated in the manuscript the assumption is that the ammonium sulphate particles are approximately spherical, and therefore the volume of a particle can be estimated using the equation for volume of a sphere. Therefore, the particle mass distribution can be estimated by multiplying the particle number distribution by the volume of a sphere and the density of ammonium sulphate. As can be seen in Figure S3 this cubic relationship between particle size and particle mass results in the ultrafine particles accounting for a large portion of the number concentration, whilst at the same time accounting for only a small percentage of the mass concentration. Therefore, in order to investigate ultrafine collection using a mass based method it was necessary to further reduce the mean size of the distribution, as was shown in Figure S3 using an impactor. The supplementary material Sect. S2.3 has been edited to better explain this.

Comment 10

Pg 17 line 4, typo, 0?

Answer 10

The authors cannot find the typo listed in the online discussion paper.

Comment 11

Section 4.1.3. What is the difference between a hydrophobic particle and an insoluble particle? Is DEHS insoluble in very dilute systems? Does DEHS remain as the original sizes generated in the droplet collection system (ie, cyclone}? The point is the one thing that is unique about this instrument is the claim that it can measure at near 100 % efficiency insoluble particles, but only one form is tested. What about collection efficiency of solid particles? Will the instrument actually collect solid particles and thus have the ability to measure ROS associated with solid particle surfaces, say for example, fresh soot particles. Are comparisons of this vortex cyclone to the Orsini et al or Peltier et al mini-cyclone valid since their test were done with truly solid particles (PSL or soot}?

Response 11

The confusion between hydrophobic and insoluble particles has been responded to in Answer 4. The authors have responded to the "uniqueness" of this system in Answer 1. The IACs purpose is to entrain particles into a solution of DMSO and the BPEAnit probe in order for reaction between the probe and sample to occur. As a steam collection device, the collection efficiency is determined by: whether or sample particles form liquid droplets inside the growth chamber; and what the collection efficiency of grown droplets is. The ammonium sulphate mass collection experiments addressed the question of the collection efficiency of grown particles. The results presented in the manuscript show that the mass of ammonium sulphate collected into the sample liquid was within error of unity with the sample aerosol mass for both fine and ultrafine mass concentration. The DEHS experiments address the question of droplet formation. DEHS was selected not because it is insoluble in the DMSO collection liquid, but because hydrophobic DEHS particles are difficult to grow into water droplets. By showing that the collection efficiency for DEHS has a cut off size of < 20 nm, it is evident that the supersaturation achieved inside the chamber is sufficiently high to grow even very small hydrophobic particles. Whether the particles are solid or liquid phase does not

influence this result. The combination of these two experimental results shows that PM is collected into the capture solution regardless of size or chemical composition. While no direct measurements of solid insoluble particles collected into the liquid stream were performed, there is no mechanism that would somehow prevent these particles from being collected in the same manner as those tested. If the purpose of the IAC were to perform direct measurements on solid particles collected into the liquid, deposition of solid particles in the liquid flow path would be the only necessary effect not considered in this characterization. However, the IAC system was developed specifically for use with the BPEAnit probe in the PINQ system. The vortex collector ensures that the particles are collected directly into the liquid rather than impaction onto a plate where they could potentially adhere and not fully react with the probe. Once collected into the liquid the reaction between the probe and sample particles is diffusion limited. This means that any potential deposition of solid particles in the liquid line will not influence the final ROS measurement as reaction with the probe has already taken place. Comparability between the IAC and the Orsini et al (2008) and Peltier et al (2007) systems would only be limited through differences in line losses in the instruments and the sensitivity of the detection methods. The efforts made to limit liquid residence time and volume would minimize this impact in the PINQ system. The updated manuscript highlights this consideration in the comparison between the different collectors.

[Figure]

**Supplement:**

[revised manuscript text omitted]

---

## Author Response (AR2)

**Associate Editor Decision**

**Comments to the Author:**

The initial reviews were both generally good, with one reviewer suggesting minor revisions and the other seeking further major revisions. The more critical reviewer considers the work similar to previously reported work. Additionally, the manuscript is criticized for not clearly distinguishing between particle bound ROS and aerosol species that produce ROS in vivo. Both of these aspects have been improved. Regarding these criticisms, the work is novel because it has a different application than the prior work using a PILS / cyclone system. Here, the purpose of the aerosol collector is to react the PM with BPEAnit, which allows for detection of reactive oxygen species, while the past work using a similar collector was focused on detection of organics or nucleic acids. The prior work regarding PM collection was partially cited and is now fully cited. Therefore, I consider the point of similarity to prior aerosol collection resolved.

With regard to ROS / OP, the text is now more clear with regard to particle bound ROS versus producing ROS in vivo. The new title, which uses "particle-bound ROS" takes the reviwer's main point. Note also that this title and the text is now in agreement with the recent paper by Puthussery et al. (2018). The reviewer takes the DTT assay as an accurate measure of OP and cites papers indicating that there are health correlations to the DTT assay. However, it is also true that particle-bound ROS would contribute to oxidative potential and thus are of interest. The authors make a good argument that development of multiple probes are useful for moving forward this emerging field and discusses relative merits of various probes fairly extensively.

From these considerations, my recommendation is that the article be accepted subject to minor revisions. Although it was not published at the time of submission of this article, the manuscript should include in Table 1 reference to the recent Puthussery et al. (2018) publication. This publication should also be included in the text in appropriate places.

**Author Response:**

The authors of this manuscript thank the Associate Editor for his professionalism, consideration and time dedicated to the revision of this manuscript. As per suggestion, we have included the recent Puthussery et al. (2018) in Table 1 and its caption, along with in: Section 1.3 "Online Collection Techniques" at Pg 3 Lines 1 and 5; Section 1.3.2 "Particle Collectors" at Pg 5 Line 31; and Section 6. "References" Pg 23 Lines 22-24.